# The effect of successful educational actions in transition from primary to secondary school

**Esther Roca**[1]*, **Pilar Fernández**[2], **Maria B. Troya**[3], **Ainhoa Flecha**[3]

1 Department of Comparative Education and History of Education, University of Valencia, Valencia, Valencian Community, Spain, 2 Department of Educational Sciences, University of Oviedo, Oviedo, Asturias, Spain, 3 Department of Sociology, Autonomous University of Barcelona, Barcelona, Catalonia, Spain

* pr.refuge.ed@uab.cat

**Data Availability Statement:** All the selective transcriptions of the interviews and communicative daily life stories are available from the Zenodo database (https://doi.org/10.5281/zenodo.10726475).

## Abstract

While data show improvement in terms of educational access, dropout rates are significant in many countries. In Spain, 28% of students drop out of school without finishing secondary school, more than double the EU average. Thus, extensive research has shown the consequences of the dropout phenomenon, including negative effects on employment, welfare dependency, as well as health and emotional problems. The transition from primary to secondary education is a critical turning point. This situation worsens in the case of refugee and migrant minors who are refugees or with migrant backgrounds. Although there is strong evidence revealing the potential of SEAs to advance educational success for all in different contexts, no research has yet explored the effect of SEAs on enabling a successful transition from primary to secondary education, which could considerably impact decreasing dropout rates. Drawing on a qualitative case study of a secondary educational centre in Spain, this study analyses the impact of the implementation of three SEAs in key aspects related to primary-secondary transitions. **Findings show how the systematic implementation of SEAs impacts the ease of the primary-secondary transition: positive relationships between the educational community are promoted, school connectedness across transition is strengthened, and academic support networks are enhanced.** The study therefore suggests the benefits of SEAs as strategies that can enhance positive primary-secondary school transitions in terms of students' perception of belongingness, and academic performance.

## Introduction

Data shows an improvement in terms of access to education; however, dropout rates are significant in many countries. In 2018, the primary school enrolment rate stood at 89% worldwide [1]. In Spain, 94.9% of 3-year-olds children attended pre-primary education [2]; nevertheless, 9.7% of 18-24-year-olds in the EU had completed at most a lower secondary education and were not in further education. In this article, we refer to secondary education as the one that includes middle and high school. Even though in Spain this figure has steadily decreased since 2012, the current figure of 11,2% has increased by 0.6% compared to 2021 and remains above the European average [3].

**Funding:** The fieldwork was done within the framework of the H2020 EU-funded project REFUGE-ED Effective practices in education, mental health and psychosocial support for the integration of refugee children (2021-2023). The funders had no role in study design, data collection and analysis, decision to publish, or preparation of the manuscript.

**Competing interests:** The authors have declared that no competing interests exist.

A wide body of research evidences the extent of the dropout phenomenon, provoking meaningful consequences, including negative effects on employment, lifetime earnings, welfare dependency, as well as health and emotional problems [4–6]. This situation worsens in the case of children and adolescents who are refugees or have migrant backgrounds. The differences between this group and host learners persist. As reported in the *'Staying the course'- The challenges facing refugee education* report, two thirds of refugee youth might never get to secondary school [7]. Moreover, this critical gap in refugee education is exacerbated when it comes to girls. Adolescent girls living in refugee camps or displaced in urban areas are less than half as likely as boys to attend and complete secondary school [7].

Even though the school drop-out occurs in most cases at the end of compulsory education, it is necessary to highlight that in some cases challenging transitions from primary to secondary school are a potential threat with consequences on **educational attainment and well-being that may lead to possible school drop-out.** Thus, the transition from primary to secondary school is regarded as one of the most difficult periods in pupils' educational careers [8]. Facing new relationships with teachers and peers, sometimes loss of friendships, a new school organization system, and a move from a smaller and personal space to a larger, complex, and impersonal one, are some of the struggles pupils must face, which might lead to a loss of sense and purpose regarding their school education. Additionally, some studies even show how this transition affects students' increased sedentary lifestyles and dietary behaviour [9]. For migrant and minority students, school segregation might involve changing neighbourhoods, separation from their community, and educational lags due to the low quality of education at primary schools.

The key question is: what conditions do students need to successfully transition from primary to secondary school so that they can contribute to a decrease in dropout rates? Furthermore, what can schools and teachers do to ensure these conditions?

Despite the importance of achieving a smooth transition between these two educational phases, these questions remain unanswered, especially because the evidence is inconsistent and shows a dependence on the characteristics of the context. For instance, a study suggests that the individual backgrounds of the students are key factors, as well as the socioeconomic composition of the neighbourhoods. This can impact the conditions of the transition and the inequalities related to education [10].

Although there are several studies (14 literature reviews published in the last 20 years) about the phenomenon of the transition from primary-secondary, there has been limited analysis of the ontology of transition [11]. Most of them focus on the relationship with teachers and peers, on the changes in relationships during the transition and on the wellbeing of students. However, there is a difficulty in predicting the impact of different factors on the real experience of the transition due to the lack of literature [12]. In addition, there are some important contributions related to the perceptions of the children and what they report about the transition, concluding that there is a relationship between the actors in the school, the children, and their parents [13].

More specifically, research on the keys that contribute to a successful primary-secondary transition in the Spanish context has not had a prominent role.

Meanwhile, solid evidence reveals the potential of Successful Educational Actions (SEAs) to advance educational success for all in primary school across diverse contexts. SEAs have been shown to improve attainment for all the students, coexistence, and the development of democratic and inclusive learning environments, as well as participation of families and the community and social cohesion [14–16], to name a few. Nevertheless, no research has yet explored the effect of these practices when implemented in secondary education, and more specifically, their effect on enabling a successful transition from primary to secondary education, especially

for students from vulnerable groups, such as those of ethnic minority and migrant backgrounds, those from single-parent families, and those with disabilities like autism spectrum conditions.

Drawing on a qualitative case study of a secondary educational centre located in Oviedo, Spain, conducted in the H2020 EU-funded project REFUGE-ED Effective practices in education, mental health, and psychosocial support for the integration of refugee children (2021–2023), which is characterized by having a highly diverse body of students (in terms of socioeconomic status and cultural and ethnic background composition), this study presents new data to shed light on the impact of the implementation of three SEAs: Interactive Groups, Dialogic Literary Gatherings, and the Extension of Learning Time in central aspects related to primary-secondary transitions. **Findings show how the systematic implementation of SEAs in the first and second grades of secondary school impacts the ease of primary-secondary transition.** This is evidenced by the decrease in school dropout rates, as well as the increase in school enrolment since the start of the SEAs. This article describes and analyses how these results have been achieved.

In line with the main objective of the H2020 EU-funded project REFUGE-ED, this study answers the research questions based on an equal dialogue with educators about their impressions of the impact of SEAs on the transition between primary and secondary school. In this way, this article outlines the main contributions of these evidence-based actions from the teachers' point of view to help other secondary schools create a suitable environment for this educational transition.

This work is organized as follows. First, the next section presents a brief literature review on the topic. Second, the research methods are explained. Third, the outcomes and their analysis are presented. Lastly, the fourth section discusses the findings of the research and presents the conclusions.

## Literature review

### An overview of school dropout in Spain and the challenges in the transition to secondary

Over the past two decades, the most notable achievement in education in most Western countries has been the overall improvement in enrolment at all levels of education, but especially in the early years. In accordance with the latest data released by the World Bank, the average global primary education enrolment was 89% in 2018, and secondary education enrolment was 66% [17]. This shows a clear downward trend in school enrolment as the age of students advances; as students' careers progress, the struggle to stay in the system becomes increasingly difficult. Despite the Spanish dropout rate having steadily decreased over the last 10 years, in 2021 it was 16.7% for men and 9.7% for women. It remains below the European average, which in 2021 was 11.4% for men and 7.9% for women [3]. The proportion of dropouts in Spain and in the EU is calculated on the basis of the Active Population Survey. Early dropout is the percentage of people aged 18 to 24 who have not completed secondary education and who are not attending any type of training or education in the previous four weeks prior to the interview [3]. According to the Terminology of European education and training policy, dropout is a disengagement from an education or training programme before its completion by a learner, without achieving the relevant learning or training objectives [18].

Although the risk of dropping out of school is accentuated at the end of primary education (in Spain, this age group is from 3 to 12 years old), there are risk factors during the transition from primary to secondary school that affect students' school engagement. There is a consensus among researchers as to the common features of this transition but not in terms of causes and consequences. According to Jindal-Snape et al. [12], there can be changes in school

relationships during the transition, and there is an influence of the design of the school buildings, as well as the teachers and the spaces in which they are learning [19]. In summary, the secondary school transition involves simultaneous changes in school environments, relationships, and academic expectations. Students' academic performance and mental health functioning are two of the factors highlighted in numerous research studies. Some studies suggest that mean academic achievement scores decrease significantly after an initial period of adjustment. According to some, this may be due to competence beliefs becoming more negative immediately following the junior high transition [20]. On the other hand, others indicate that not all students experience changes to the same extent, in part because of individual differences in self-esteem development [21], as well as the context in which they develop, such as those with high stakeholder involvement in transition [13], the student's socially disadvantaged background, or students with disabilities [22].

In any case, despite some discrepancies between the studies, in the last decade, sufficient evidence points out the importance of the learning environment during this process. Schools that foster positive personal relationships with teachers, positive peer interaction, and engaging learning activities for students to enjoy school tend to facilitate the transition from primary to secondary school. These characteristics remain critical to student engagement in secondary school as well [23]. In fact, van Rens et al. [13] reveal how the relationship between stakeholders, the children, and their parents can improve the transition.

Another of the factors most often mentioned to explain the origin of school dropout is the lack of interest and boredom with the school's own routines and basic pedagogical activities. According to Daschmann et al. [24], monotony, lack of meaning, insufficient challenge, and lack of student involvement are some of the precursors of academic boredom. When students feel that they do not learn and do not participate in the life and dynamics of the school, they lose their sense of education, leading to demotivation regarding their studies. It should be noted that there are also extracurricular factors related to the conditions and context of the students that are beyond the school's control, and that can also lead to dropout. Nonetheless, at this point, we want to emphasize those factors to which special attention should be paid because they can be addressed by schools.

In line with this, the commitment to quality learning for all in environments with different educational levels stands out. From this perspective, the same knowledge is made available to all students, regardless of their educational level and origin. In contrast to segregationist policies and programmes that have not proven to impact the inclusion of less-advantaged students, this approach is focused on offering everyone the same opportunities for improvement. To this end, some propose extracurricular actions to enable students with learning difficulties to catch up. However, the I-ED project later demonstrated that these inclusive actions can also be carried out during school hours and bring results in the improvement of learning and school coexistence. The answer lies in appropriately organized and resourced heterogeneous classrooms; therefore, students with disabilities achieve better academic results and have a better self-concept compared to those in segregated classrooms [25]. In this way, I-ED identified successful actions that contribute to overcoming school failure: heterogeneous grouping with reallocation of existing human resources, extension of learning time, and certain types of family and community education.

## Worsening primary-secondary transitions: Challenges for migrant and refugee students

When it comes to students from at-risk groups (those of low socioeconomic status, cultural and ethnic minorities, immigrants, or those with mental or physical disabilities, among others), the situation is even more difficult. In the case of refugee and immigrant children and

youth, the enrolment rate is lower than that of host students. According to the UNHCR Education Report 2021, worldwide, the enrolment of refugees in primary education is 68%, and in secondary education is 34% [7]. In addition, the gross secondary schooling rate of refugees in 41 reporting countries stood at 34%, compared to their host counterparts, which was significantly higher [26]. Across all OECD countries, 7% of native-born pupils with immigrant backgrounds dropped out of school prematurely, and in the EU, 9%. Conversely, the dropout levels of students who were foreign-born were 15% in the EU and 11% across the OECD. In Spain, the shares exceeded 13%, and in some countries like Switzerland, Austria, and Slovenia, young people with a non-EU background are more than twice as likely to drop out [2].

This shows, on the one hand, the disparities between students with migrant backgrounds and host students in primary and secondary school and, on the other hand, once again illustrates that enrolment in secondary school is lower than in primary school. This reflects the difficulty for students to remain in the educational system as they get older.

In Spain, the national education statistics do not capture refugee and migrant children. However, 95% of the 6,200 refugee and migrant children were enrolled in secondary school in the 2017–2018 school year [27]. Also, Spain is one of the countries with a higher gap in educational attainment between non-migrants and migrants. The share of migrant young adults aged 18–24 with educational attainment at levels 0–2 (less than primary, primary, and lower secondary education) was 40.8% in 2019. And the share of non-migrants was 39.4%.

The latest data for 2021 show a considerable difference between nationals and migrants in terms of early school dropouts in Europe. Thus, 8% of nationals between 18 and 24 have completed no more than lower secondary education and are not involved in further education or training; in comparison with 23% of EU mobile citizens and 26% of non-EU citizens [28]. According to the same data, this marked disparity is also evident in the analysis of the population by level of education: 45% of non-EU citizens have a low level of education, meaning that they have completed less than primary, primary, and lower secondary education. In contrast to 19% of nationals and 27% of EU mobile citizens.

Based on this premise, the transition to secondary school for refugee children and youth represents an even greater challenge. In addition to the aspects mentioned in the previous section, most refugee and migrant pupils come from environments fraught with conflict and even violence. In many cases, they are separated from their families and the environment they know, moving to territories with a different culture and language, which further deepens the challenges involved in this educational transition. A study conducted in Amsterdam concludes that school advice is highly differentiated among children from different migrant and socio-economic backgrounds. Furthermore, it suggests that while most of these educational inequalities may be explained by individual characteristics, residential and school segregation intensify them. Despite the undeniable effect of individual factors on transitions to secondary school, the study suggests that both the neighbourhood and the primary school context appear to have a direct effect on a child's educational level during this period [10].

As this article aims to demonstrate, there is a large body of research on the transition from primary to secondary school, but much remains to be done regarding immigrant and, more specifically, refugee children and youth during this phase. Special attention should be paid to the increasing human mobility, which is reflected in the composition of the world's population. Based on data from UNHCR, at the end of 2021, there were 27.1 million refugees and 53.2 million internally displaced persons (due to conflict and violence) around the world, and overall, EU countries granted protection to around 275,000 people in 2021 [26]. This reflects the urgency of putting in place mechanisms to bridge the disparities between children of refugee and immigrant origin and the rest, as well as expanding evidence-based research that has a social impact on these vulnerable groups.

## Factors to consider for a positive transition

Some of the most obvious results of a successful transition from primary to secondary school are lower dropout rates. With this in mind, it is important to understand that there are out-of-school and in-school factors that explain dropout. The former has more to do with the context and situation of the students (e.g. socioeconomic status). Although these factors must be addressed by the state, they cannot be changed by schools. In-school factors, on the other hand, are strictly linked to the way schools and teachers operate in the classroom. For example, teacher authoritarianism or lack of student interest may be in-school factors influencing some students to drop out. Although school dropout is explained by a variety of circumstances, it is necessary to point out that the actions promoted by schools are crucial to prevent students from dropping out of school.

Providing quality education for all students, without distinction of any kind, as well as promoting violence-free spaces, are key aspects that make a difference. Extensive research [8, 29, 30] points out that friendship is a major source of concern for students transitioning to secondary school. For this reason, having good relationships with peers contributes to students' well-being and sense of belonging in school. In this sense, there are several research studies that show how children's experiences can be influenced positively or negatively depending on their relationships with peers and teachers [12]. Therefore, the transition is shaped by teachers and parents, and it is important to consider all these perspectives to analyse the transition of the pupils [13, 31]. In fact, family support can enhance this transition to the extent that children benefit when schools and parents are on the same page, so that both teachers and parents have the opportunity to help children develop their social and emotional skills [32]. In line with these studies, Francés et al. [33] show the importance not only of school-student-family communication but also the essential role of mentoring in the transition.

Therefore, to achieve a successful transition from primary to secondary school, attention must be paid to school engagement and family involvement, as evidenced by students' sense of belonging and well-being.

## Working on the shoulders of giants: Effective practices based on evidence

Successful Educational Actions (SEAs) were conceived as a result of INCLUD-ED, a research project led by the CREA research centre from 2006 to 2011 whose purpose was to find strategies that contribute to inclusion and social cohesion in Europe in the educational field [34]. The theoretical basis of the model is grounded in Dialogic Learning, a concept that places "the dialogicity of the person at the centre of the teaching and learning processes" [35]. Additionally, its foundation is rooted in two key factors for learning in today's society: interactions and community involvement [36]. At this point, involving families and communities in the educational process and in decision-making, creating meaning, and making the most of the cultural intelligence of all are some of the main features of SEAs.

Thus, SEAs seek to improve learning and school coexistence through comprehensive transformation, both inside and outside the classroom. These practices have been shown, with scientific evidence, to reverse problems of exclusion and inequality, regardless of the context in which they are implemented [37]. There are currently more than 3,000 Learning Communities worldwide and more than 10,000 educational centres that implement one or more of the SEAs.

As mentioned above, its social impact has crossed borders. Numerous studies have shown positive results regardless of the context [38–41]. In 2011, converting schools into Learning Communities was even proposed as one of the recommendations made by the European Commission to reduce school dropout rates [42]. Similarly, the results have been demonstrated through various European projects that have even had an impact on public policy in some

countries, such as SEAS4ALL, EnlargeSEAS, STEP4SEAS, and Learning Communities in Portugal. In addition, since 2013, the SEAs have been implemented in educational centres in Latin America, with students from intercultural, migrant, and refugee backgrounds.

There are six Successful Educational Actions. Nonetheless, this study will analyse the impact of four of them that are implemented in the school: Dialogic Gatherings, Interactive Groups, the Dialogic Model of Prevention and Resolution of Conflicts, and the Extension of Learning Time, which are briefly explained below.

- Dialogic Gatherings: is an activity of reading and joint construction of knowledge, based on the reflection of universal classics. In addition to improving oral expression, it increases vocabulary, comprehension, analysis from different perspectives and creates meaning through the link between reading and the experiences of the participants.

- Interactive Groups: is a form of classroom organisation that currently gives the best results in terms of improving learning and coexistence. Through interactive groups, interactions are multiplied and diversified, while effective working time is increased. They are characterised by the fact that they are an organisation that includes the pupils, with the help of more adults in addition to the teacher in charge of the classroom.

- Dialogic Model of Prevention and Resolution of Conflicts: his is a strategy to improve coexistence by overcoming inequalities through egalitarian dialogue. It proposes consensus as the main mechanism for conflict resolution, which requires the participation of the whole community. As part of the model, the development of a coexistence rule with the whole educational community is proposed.

- Extension of Learning Time: after-school programmes such as homework clubs or tutored libraries are organised. These are also characterised by the support of volunteers.

On this basis, the H2020 EU-funded project REFUGE-Ed, an international collaborative research project funded by the European Commission's Horizon 2020 framework program, seeks to improve the academic performance and dynamic integration of migrant, refugee, and asylum-seeking children through effective practices in education (Successful Educational Actions) as well as mental health and psychosocial support. The case study for this research involves one of the project's pilots, located in Oviedo, Spain.

## Methods

The study revolves around the following research question: To what extent, if any, can the implementation of SEAs contribute to a more successful transition from primary to secondary school? To this end, the study proposes interviews with teachers and students from the centre. This way, the question is answered from the point of view of the educators and collaborators in the centre and students through spaces for reflection and equal dialogue with the research team.

The data used in this article were collected in the framework of the *H2020 project REFU-GE-ED effective practices in education, mental health, and psychosocial support for the integration of refugee children* from 2021 to 2023. The REFUGE-ED project brings together two fields of expertise: education and mental health and psychosocial support (MHPSS) in humanitarian settings to improve academic achievement and the dynamic integration of migrant, refugee, and asylum-seeking children [43].

### Data collection plan

For this specific study, 15 interviews with teachers and students were conducted, and documents and records on school coexistence, permanence, retention, and academic improvement

**Table 1. Profiles of interviewees.**

| Time in the centre/Gender | Female | Male |
|---|---|---|
| 1–3 years in the centre | 4 | 2 |
| 4–6 years in the centre | 1 | 1 |
| 7 or more years in the centre | 1 | 1 |

collected by the school were reviewed. A purposive sample was used because the researchers had a clear idea of the characteristics, they were interested in exploring and wanted to select a sample representative of those characteristics, depending on the objective of the study.

Therefore, five students from different grades were interviewed, with special emphasis on the 1st year of secondary school, and with migrant or refugee backgrounds. For the selection of the participants, consideration was also given to their life stories, transformed in the centre.

Additionally, 10 interviews were conducted with the school's teaching staff to understand the role played by the implementation of the Successful Educational Actions in the centre in improving the conditions for a smoother transition from primary to secondary school. For the teacher interviews, the researchers considered that the teachers were implementing SEAs or had done so in previous years, and that there was diversity in terms of years of experience, area of specialization, and gender. Table 1 summarises the profiles of the teachers interviewed.

The interviews were conducted using the communicative methodology, which is based on egalitarian dialogue between the researchers and the subjects of study [44]. This approach "contributes to overcoming situations of inequality and promotes the inclusion of less favoured social groups" [45]. In this way, both parties are committed to sharing their knowledge to reflect and better comprehend the realities analysed. The interviews had an average duration of 1 hour, during which participants were asked how they perceive changes in coexistence and academics since they began participating in the SEAs.

## Data analysis

In pursuing the purpose of the study, three main categories were established to guide the work of the H2020 EU-funded project REFUGE-ED: well-being, sense of belonging, and engagement at school. These three key points guarantee the inclusion of less-favoured groups; in this case, migrants, and refugees. Table 2 describes the categories and subcategories used in the study.

Once the fieldwork was completed, the content was analysed using two dimensions proposed by the Communicative Methodology: the exclusory and transformative dimensions. The former refers to all the barriers that prevent students from successfully transitioning from primary to secondary school. In contrast, the transformative dimension identifies how those barriers are overcome.

"*This definition of both exclusory and transformative dimensions enables the identification of existing alternatives that can improve social situations and overcome the social problems*

**Table 2. Coding scheme.**

| Analytical category | Wellbeing | Sense of belonging | Academic performance |
|---|---|---|---|
| **Subcategories** | -Positive relationships | -School attendance | -improvements in the teaching-learning process |
| | -Conflict prevention and resolution | -School engagement | |
| | -Expectations | -Community participation | |

*studied. Additionally, this analysis includes the voices of social actors, so that understanding what is exclusionary and what is transformative is based on people's own knowledge. This enables a better understanding of the phenomena studied and supports more effective processes of change"* [46].

Data collection took place between January and May 2023 through qualitative fieldwork and document research. To protect the data of the persons involved in this study, pseudonyms are used in this document.

## Presentation of the case study

The research takes as a case study the secondary school IES Alfonso II, located in Oviedo, Spain, according to the following criteria: 1) It is a large school, with a highly diverse student body (there are almost a thousand students in the centre, 90 teachers, and 14 other staff members). The school offers different types of studies: secondary education, adult education, different post-mandatory education courses, and night courses. 2) Since 2019, the school has been implementing Successful Educational Actions. 3) In 2021, the school was included as one of the pilots of the H2020 REFUGE-ED project. 4) Since the implementation of the SEAs, the school has shown a profound transformation, both in terms of coexistence and in the improvement of academic success. Because of the improvement in results at all levels, this article analyses how this has been achieved. In Fig 1 there are some data that reflect this process of comprehensive change and improvement:

The data shown in Fig 1 also demonstrates how this increase in expectations translates into some institutional indicators since the SEAs started in the centre. As the data indicates, in the first year of implementation of SEAs (2019–2020), the number of cases of students referred to Social Services decreased significantly from the previous year, i.e. from 28 to 7, a trend that continued until the 2020–2021 academic year. It should be noted that cases are referred to Social Services when a child is seriously absent from school (more than 20% of absences during school hours). Subsequently, in the 2021–2022 school year, there is a spike in these cases, rising

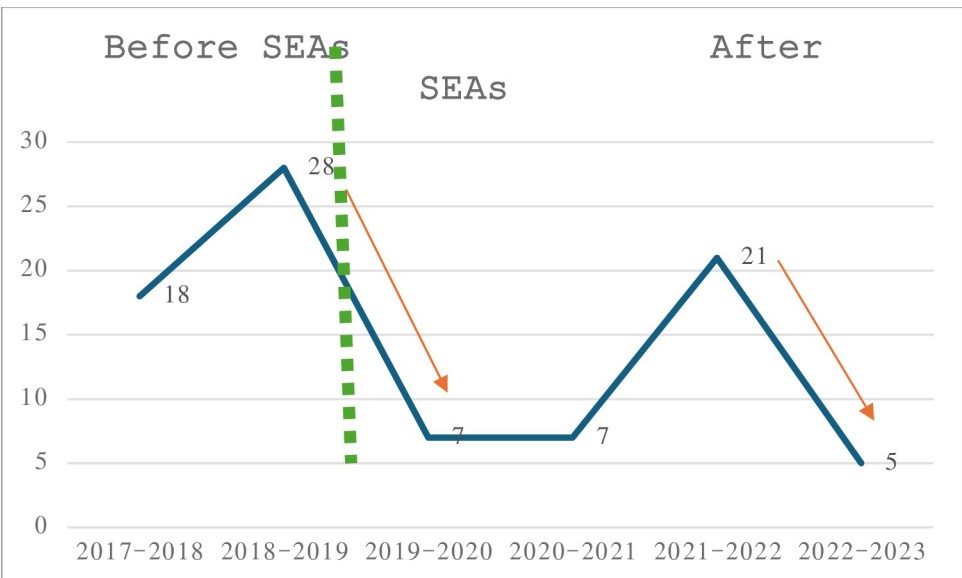

**Fig 1. Reduction of absenteeism.** Source: data provided by the head of studies of the centre.

to 21 (the causes of this phenomenon have not been studied so far). Nevertheless, in the last year, the number of cases decreases again to 5, the lowest number since 2017–2018.

It is worth mentioning that the cases corresponding to the period 2019–2020 were reported only until March, when the school was closed due to the pandemic situation, and classes were switched to telematic learning. In this context, both the dialogic gatherings and the tutored library continued online. Another important aspect to highlight is that, unlike most Spanish public schools, which alternated between face-to-face and online classes to avoid overcrowding, IES Alfonso II maintained face-to-face classes for all students.

Additionally, the data shows a positive impact on the increase in enrolment since the SEAs started. Traditionally, IES Alfonso II did not fill all the available school places at the beginning of the school year. However, for the academic year 2023–2024, 100% of the places for the first year have been reserved, and there is even a waiting list for students from non-affiliated schools. Also, the Department of Education has granted the school an extra group in addition to the 4 existing ones for the third year due to the high demand for the centre from families. This data reflects the positive perception that families have of the work being done at the centre, especially since these practices involving the whole community have been carried out.

Another noticeable result since the SEAs started at the centre is the decrease in conflicts. As shown in Table 1, there is a considerable decrease in the number of sanctioned students in the 2022–2023 school year compared to 2020–2021. The difference is 37 students. Furthermore, this reduction is also evident in the recidivism of negative behaviour. While in 2020–2021, 26 students had between 3 and 10 sanctions, in 2022–2023 there were only 9. In other words, as shown in Table 3, there is a reduction of more than 65% in recurrent students with more than 3 sanctions.

Another key point to note is that, according to the school's administrative body, since the SEAs have been systematically implemented, dropout rates have decreased, and the academic level of students has increased. At the end of the 2022–2023 school year, the number of repeaters in the first year (1st ESO) was less than half of the previous year.

As shown in Fig 2, a decrease in the number of repeaters can be observed over the four years since the beginning of the implementation of SEAs. It should be noted that the 2019–2020 school year is not included in the data because it was the year of containment and does not provide comparable data for any parameter.

However, the difference is significantly greater in the case of 1st ESO, the year in which SEAs are most frequently implemented (at least twice a week). The data is particularly relevant given that in Spain, overall re-enrolment rates have increased across all grades of secondary education from 4.2% in 2021–22 to 7.6% in 2023 [47].

In particular, the reduction of school dropout is a key criterion for the selection of this school as a case study. As the literature review has shown, early dropout remains one of the main challenges facing national education systems. In this regard, the school is at its best when the implementation of Successful Educational Actions is frequent and systematic. Moreover, their implementation has promoted the involvement of families in school. As a result, conflicts

**Table 3. Recurrence of negative behaviours in the school.**

|  | First evaluation 2020–2021 | Second evaluation 2021–2022 | Third evaluation 2022–2023 |
|---|---|---|---|
| **Sanctioned students** | 123 | 108 | 86 |
| **Students with between 3 and 10 sanctions** | 26 | 13 | 9 |

Source: data provided by the head of studies of the centre

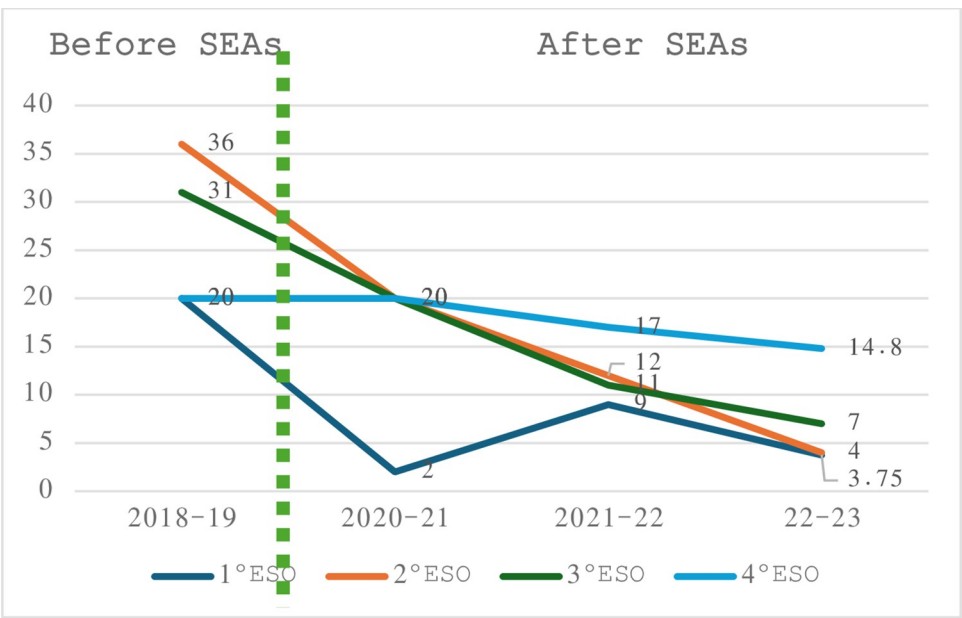

**Fig 2. Evolution of grade repetition before and after SEAs.** Source: data provided by the head of studies of the centre.

have decreased, especially repeated offenses. This also translates into increased enrolment rates as students and families see the school as a place where their voice is heard.

Considering these results, as mentioned above, this article will analyse the keys to success that have led to the improvement of these indicators.

## Relevant contextual aspects

This section includes relevant information to understand how the Spanish education system works. The Spanish education system is structured into general education and special education. General education includes early childhood education, primary education, compulsory secondary education, baccalaureate, vocational training, and university education. In 1990, the General Education Law was replaced by the General Organic Law of the Education System (LOGSE). As a result, General Basic Education was replaced by what is now known as Compulsory Secondary Education (ESO, by its Spanish acronym). Thus, ESO is the last compulsory stage and covers four school years for students aged 12 to 16.

## Findings

### Newcomers students first time at the centre has been in an atmosphere of mutual help

Drawing from the analysis of the different inputs and situations explained by the teachers interviewed, the main transformative aspects that contribute to a positive transition of students from primary to secondary school are presented. Firstly, the results show the direct contribution of Interactive Groups, Dialogic Literary Gatherings, and the Extension of Learning Time to students' quality of school life. This quality of school life is built on components such as the perception of safety, perceiving the school as a safe space, having quality relationships or even friendships based on trust and respect, maintaining positive relationships between different

members of the educational community (students, teachers, and families), as well as seeing the school as an environment where dialogue rather than violence prevails.

In terms of the development of safe spaces in the school, according to the interviewees, the Successful Educational Actions (SEAs) have had an impact on the development of safe spaces within the school. It's worth noting that these spaces are not limited to the period in which the SEAs are carried out but have expanded beyond the classroom, creating a safe environment in areas like the corridors and playground.

> "In interactive groups, as well as working as a team, they [students] feel safe and contribute their opinions freely. Children interact who in normal circumstances would not interact if you grouped them heterogeneously. When someone new in the interactive groups participates, they feel part of it, they feel they are someone and the others listen to their ideas (. . .) [The SEAs] work over time, each session creates new interactions that remain in the classroom, children who did not speak when they arrived, they create friendships, the dynamics of the classroom change, the interactive groups leave a mark when they return to the classroom, the air changes" (teacher).

The impact of the three Successful Educational Actions on which this study focuses (Dialogic Gatherings -DG-, Interactive Groups -IG-, and the Extension of Learning Time) show certain particularities that are explained throughout the analysis of the results. Specifically, the Dialogic Gatherings held at the centre have broken the fear not only of participating in class but also of talking about emotions, feelings, and opinions, and sharing personal experiences related to the book they were reading.

> "I like the gatherings (DGs) because we share our experiences. Sometimes a person needs a voice of encouragement among us. Maybe the other person is going through the same or worse situation than us", (student).

This clearly shows that the DGs are perceived by the students as a safe space where they are respected.

> "They feel supported, safe, they know that no one is going to laugh or make fun of what you say, this means that little by little they participate, this is also noticeable in the classroom, they are not afraid to express what they think, the dynamic changes, the same people don't always raise their hands (. . .) By working with Interactive Groups in the classroom it is noticeable, they raise their hands more, they are not afraid to make mistakes" (teacher).

This perception of security has even turned the centre into a community in which its members look out for each other. This aspect is particularly relevant since the centre is one of the largest in the city, which can make it intimidating at first glance for any new student. For some of them, it can even be an obstacle when adapting, especially for those with a migrant background.

Regarding the Extension of Learning Time through the Tutored Library, given its noticeable impact, the school authorities have used this action as a tool to ease the transition and adaptation process for newcomers, especially migrants. Therefore, when a new student arrives at the school, they and their families are invited to the Tutored Library before their first day of school. This way, the Tutored Library becomes their first safe space to introduce themselves, get to know other students, the teachers, and the school itself. This strategy has proven to give newcomers confidence and reduce pressure on their first day because they already know some of their classmates, staff, and teachers.

"(New students) They arrive in the evening at the tutored library (. . .) the next day (first day of classes) they go to their class, and they realise that they already know 4–5 children with whom they were in the TL and have some friendships. Their first time at the centre has been in an atmosphere of mutual help, the next day they join the group with friends", (teacher).

On the other hand, Interactive Groups seem to make it easier for newcomers or shy students to adapt. Firstly, because they work in small groups. Additionally, the aim of this practice is the co-creation of knowledge through dialogue. In this approach, students do not compete; instead, they help each other.

"In the Interactive Groups we talk more, we are looser, we are freer, because we are with people who are not teachers (referring to the volunteers), they can listen to us and understand us", (student).

At this point, the presence of a volunteer changes the dynamics of the entire class. The volunteer not only promotes interactions between students, but their presence also seems to encourage better behaviour among the students.

"(when there are volunteers in class) I feel good because I know these people. The volunteers stay there to assist and if we have a question, they help us. The volunteers are nice", (student).

## Break down stereotypes and create new healthy relationships

As mentioned above, various studies show that improved well-being has a positive impact on students' academic performance, which in turn contributes to higher expectations for their own future, helping them break the glass ceiling—a metaphor for discriminatory barriers. In other words, once students overcome the academic difficulties they face and see that they can do it, they realize that they are perfectly capable of achieving anything they dream of.

It's worth noting that expectations operate at multiple levels. On one hand, there are the students' expectations for their own future. Then, there are the teachers' expectations for their students, and finally, the family's expectations for their children. The results of this study reveal that SEAs address all three levels.

"*There is a boy from Guinea who has been here almost all his life, he has done a training cycle in physical education and today he told me that since he is a volunteer, he is now clearer about it, and he is going to sit the exams to be able to become a teacher. That is to say, it also helps the personal promotion of volunteers, migrants and non-migrants", (head of the Spanish language and literature department and coordinator of the innovation project).*

"Expectations change, if I compare it with other schools, when a pupil starts with difficulties it is very difficult to improve them, however, with the SEAs you see the children improve very quickly, when they see how they improve they are more eager to improve, it is an exponential learning process, the expectations are important, (teacher).

In this sense, SEAs work on the basis of not lowering the bar for pupils with difficulties but maintaining it and simultaneously providing spaces and interactions that benefit them in overcoming their difficulties. In doing so, students start to believe in themselves, and the learning

and opportunity gap is reduced. Moreover, teachers begin to believe in their students, and since families have the opportunity to participate in the classroom, those who volunteer realize the capacity of the children. Many of them were surprised to participate in Interactive Groups and Dialogic Gatherings. Importantly, these higher expectations were not only reflected in the students but also in the volunteers involved in the centre.

According to the teaching staff's reflection regarding improved results, as presented in the case study, they mention that since the beginning of the school year there has been a reduction in the seriousness of negative behaviors affecting good coexistence. The Head of Studies noted that, in general, negative behavior sanctions have been less severe. Additionally, since the SEAs have been implemented, students have open spaces for dialogue, and as a result, they feel safer and are more likely to report negative situations or dynamics before they escalate, allowing for preventive action.

The strong presence of egalitarian dialogue in all SEAs was highlighted in all interviews. From the seating arrangements in a circle, facing each other to avoid hierarchical positions between teachers and students, to the atmosphere of respect and tolerance that develops in the gatherings, these practices enable egalitarian relations where dialogue based on the validity of arguments, rather than power, prevails. These aspects create a fertile environment for the growth of healthy relationships and ultimately strong friendships, which helps newcomers adapt and contributes to the well-being of students at the centre.

> "In each session (Interactive Groups or DLGs) they create new interactions that remain in the classroom, children who don't talk when they arrive, they create friendships, the dynamics of the classroom change, the interactive groups leave a mark when they return to the classroom, the air changes", (teacher)

## Sense of belonging as a booster for academic success

From its definition, a sense of belonging means feeling comfortable and part of an environment. As shown in the analysis in the previous sections, the different dimensions described above are interconnected. To cultivate a sense of belonging at school, it is desirable to create a positive and supportive school environment, foster social interactions, celebrate diversity, encourage student participation, foster relationships between teachers, students, and families, and promote rising expectations. In other words, the components explained previously in this article.

Therefore, in this section, the analysis will mainly focus on the sense of unity that is promoted in the school, emphasizing the impact of community involvement in the whole teaching-learning process and its impact outside the classroom, such as in the lives of family members. Finally, the analysis will explore how this effect can contribute positively to improved academic performance.

One aspect highlighted in the interviews is the positive perception of family involvement through the Successful Educational Actions, not only by teachers who are implementing them but also by those who have chosen not to participate in them.

> Our first African volunteer was coming down the corridor and someone (teacher) who is not involved in the SEAs said, "it's good that you're coming, you're doing a lot of good for our children", (teacher)

This means that the transformation brought about by SEAs in community involvement in the teaching-learning process and in school decision-making has transcended the time and

space in which these practices take place. Moreover, this transformation seems to extend into the lives of those who participate in the SEAs. According to the interviewees, the volunteers' motivation to participate in the centre is undeniable. For example, a story was shared about a mother who wishes to remain involved despite her difficult health situation:

> "This year we have a couple of cases of mothers who are sick, in a serious health situation, and they come whenever they can. I told one of them that she doesn't need to come so much because she has to take care of herself and she replied " no, because this gives me life", (head of the Spanish language and literature department and coordinator of the innovation project, IES Alfonso II).

On the other hand, one aspect noted in the interviews was that, although the community participates in the SEAs and other areas of the school, there is generally a lack of participation from the families of students with an immigrant or refugee background. This may be due to their economic and legal situations. At this point, both teachers and school staff consider the participation of these families to be essential to promote the integration of their children and improve their academic performance.

## Success is for everyone

The interviewees expressed a direct contribution of the Successful Educational Actions to the learning process as a whole. Considering that the learning process involves different cognitive skills such as attention, language, processing, and organisation, as well as social interaction skills, SEAs demonstrate a contribution in all these areas.

According to teachers, the different types of Dialogic Gatherings (DG) held at the centre— Literary, Feminist, and Scientific—create an environment where students feel motivated because they can relate the reading to their personal experiences, which leads to the development of higher-order thinking. Additionally, by creating a different classroom climate, students tend to pay attention because the focus is not solely on the teacher throughout the entire class; their role shifts from passive to active.

Regarding the Interactive Groups, the teachers stated that these allow them to cover more than one exercise in one hour, focusing on different skills and knowledge. In contrast, in a traditional class, this would take more than an hour of class time.

> "They (students) love it (Interactive Groups), it's something that is difficult to explain, to do the number of exercises you do without interactive groups you need three sessions and in interactive groups they do everything, they say it themselves "even if I don't want to, I learn, I can't disconnect". Everyone works at the same time", (teacher)

Certainly, when the volunteers are family members of the students, the impact is multiplied. Family participation in the classroom helps students in their process of adaptation and shapes their expectations for their own future. Moreover, the time spent in the classroom allows family members to gain a new understanding of the value of the work that teachers do. This principle applies to all three SEAs—DLGs, Interactive Groups, and the Extension of Learning Time through the Tutored Library.

> "I can tell you the case of a Ukrainian girl who speaks Spanish but had difficulties. The fact that she comes to Tutored Library with her mother makes her learn the language very well, she is a different girl, she understands everything and before it was very difficult, the fact that she comes with her mother who speaks Spanish is wonderful. When the families work

with them, the children try to learn more, the dynamics between them also works, they know that the families care and they know that we talk to their families, that creates a very good atmosphere, to study and learn", (teacher).

Another aspect that was highlighted in the interviews is the improvement in engagement in school activities, which may be a contributing factor to the decrease in the dropout rate at the centre.

"I thought it (the tutored library) was very good because it is very much needed here and more so for people coming from abroad, the content (in Spain) is different to that of our countries. And coming here and having it explained to you is very good because they help you a lot", (student).

"I went last year to the tutored library; the head of studies recommended me to go over the subjects and it went very well. I went every day; I met a lot of people. At the beginning I was very scared and embarrassed, and I found it very impressive how they (volunteers) look after the wellbeing of the students. The teachers do their bit and volunteer in these classes in the afternoon, when they could be at home, and they are here, sharing with us", (student).

This could be explained by the fact that once students realize they can succeed, they become motivated and committed to school.

"Successful Educational Actions are a way of enriching students' education, and avoiding drop-out is a consequence", (teacher).

"(The SEAs) are closely linked to the fact that fewer children drop out, the impact is positive, the dropout rate has gone down, even this year", (head of studies).

As a final remark, according to the interviewees, thanks to the SEAs, the transition is less difficult, largely because there are many diverse spaces for participation and interaction. Moreover, they note that the fact that students are in safe environments with their families makes the transition easier.

"I have heard from parents who say that the children were very afraid to come and that after the first week they feel at ease, I connect this with the performances, with the tutored library, the interactive groups and the dialogic gatherings", (teacher).

## Discussion

The findings of this study indicate that there is a positive contribution of Successful Educational Actions (SEAs) in central aspects related to the transition from primary to secondary school, including well-being, sense of belonging, and academic success. The creation of safe spaces through these effective actions was found to increase school engagement, fellowship among pupils, and ultimately the overcoming of challenges like language acquisition, especially for children with migrant backgrounds. Consequently, the school becomes an environment suitable for learning and socialisation among peers. The findings of this study revealed that once these SEAs are systematically implemented, the results are enhanced, including a considerable reduction in conflicts within the centre and even a reduction in the drop-out rate.

In line with previous research, the transition to secondary education involves a period of adaptation. At this point, the importance of the learning environment during this process is irrefutable. This study shows how the frequent and systematic implementation of SEAs allows for a smooth transition between primary and secondary school through the creation of safe spaces that transform the school into a true learning environment. The foundation of these actions is based on egalitarian dialogue and interaction between all parties (i.e., children, teachers, community, parents, and family). As suggested by van Rens (2018), the relationship between stakeholders, children, and their parents can improve transition, which has been demonstrated throughout this study.

While some studies have pointed to the challenges faced by children from vulnerable groups such as migrants and refugees (UNHCR, 2021), as well as boredom and lack of expectations (Daschmann et al., 2011) as reasons why children drop out of school, this study showed how these obstacles are being overcome and turned into opportunities. According to the SEAs approach, the same knowledge is made available to all students, regardless of their educational level and origin, hence providing equal opportunities for improvement to everyone.

On the other hand, the study reveals how companionship and even friendship is fostered through SEAs, which has been shown to be a major concern for students transitioning to secondary education (Pratt & George, 2005; Rice et al., 2011; Zeedyk et al., 2003).

As this article has demonstrated, while previous research has focused on the causes and consequences of early school leaving and on the difficulties of transition from primary to secondary education, these results outline concrete lines of action applied in a heterogenous context, which can help other schools reverse this phenomenon.

## Conclusions

According to the results of this study, there is evidence of a positive contribution of Successful Educational Actions (SEAs) in central aspects related to the primary-to-secondary transition, including well-being, sense of belonging, and academic success. The teachers stressed that SEAs have helped create safe spaces within the school, which have expanded throughout the school over time, especially for those in vulnerable situations, contributing to an overall improvement in school coexistence. Research reveals that these environments are stimulating because they are supportive of each other. As a result, conflict is significantly reduced, and violence and discrimination are not tolerated. Moreover, diversity is seen as a positive aspect, making the school an inclusive space where egalitarian dialogue prevails.

As part of this improvement, the contribution of SEAs has also been noted in strengthening existing relationships between students and creating environments where healthy relationships and new friendships flourish. In this sense, the change in narrative regarding diversity within the school has been key. Instead of being seen as a threat, diversity is viewed as an opportunity for enrichment.

Another aspect facilitating an easier transition from primary to secondary school is related to the sense of belonging in the new school. In fact, the results show that the SEAs implemented in the school have improved pupils' perception of the school. This improvement is also linked to elevated expectations at all levels (students about their own future, teachers about their pupils' future, and families about their children's future).

The results of the fieldwork reveal the importance of community participation in the learning process of students. However, one of the weaknesses detected was the lack of involvement from the families of the most vulnerable students, especially those with an immigrant or refugee background. This may be due to their socio-economic or legal status. Given that family involvement is key to facilitating the transition from primary to secondary school, attention should be paid to finding different ways to address this difficulty.

Based on the findings of this article, there is evidence of a clear influence of SEAs in enhancing students' academic success, measured not only in terms of improved knowledge and academic achievement but also in terms of motivation and school engagement.

It is worth noting that, according to the interviews and school indicators, the frequency and systematic implementation of SEAs are two main factors influencing the results. In other words, the positive outcomes are due to a systematic implementation of these actions, rather than sporadic application. This is evident because the impact is greater in the first course (1st ESO), a year in which the frequency of SEAs has been intentionally increased to facilitate the transition to secondary school.

The results of this study shed light on how the transition process between primary and secondary school can be facilitated, especially for immigrant and refugee students, through the application of evidence-based practices in different contexts and settings.

## Acknowledgments

The authors would like to thank IES Alfonso de Oviedo-Spain for their active participation and involvement in the research as well as all the people who participated in the fieldwork.

## Author Contributions

**Conceptualization:** Esther Roca.

**Data curation:** Pilar Fernández, Maria B. Troya.

**Investigation:** Esther Roca, Pilar Fernández, Maria B. Troya, Ainhoa Flecha.

**Methodology:** Esther Roca, Maria B. Troya, Ainhoa Flecha.

**Resources:** Ainhoa Flecha.

**Supervision:** Esther Roca.

**Validation:** Esther Roca.

**Writing – original draft:** Pilar Fernández, Maria B. Troya, Ainhoa Flecha.

**Writing – review & editing:** Esther Roca, Maria B. Troya.

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
