## [Decision Letter · Decision Letter 0]

18 Jan 2024

PONE-D-23-38228The effect of Successful Educational Actions in educational transition from primary to secondary schoolPLOS ONE

Dear Dr. Roca

Thank you for submitting your manuscript to PLOS ONE. After careful consideration, we feel that it has merit but does not fully meet PLOS ONE’s publication criteria as it currently stands. Therefore, we invite you to submit a revised version of the manuscript that addresses the points raised during the review process.

Kind regards,

Chinaza Uleanya

Academic Editor

PLOS ONE

Journal Requirements:

The fieldwork was done within the framework of the H2020 EU-funded project REFUGE-ED Effective practices in education, mental health and psychosocial support for the integration of refugee children (2021-2023).

5. Please amend either the title on the online submission form (via Edit Submission) or the title in the manuscript so that they are identical.

Additional Editor Comments:

Thank you for your submission. Following the report of the reviewers, the paper requires minor revision. Please, attend to the comments of the reviewers.

Reviewers' comments:

Reviewer's Responses to Questions

**Comments to the Author**

1. Is the manuscript technically sound, and do the data support the conclusions?

Reviewer #1: Yes

Reviewer #2: Yes

2. Has the statistical analysis been performed appropriately and rigorously? 

Reviewer #1: N/A

Reviewer #2: Yes

3. Have the authors made all data underlying the findings in their manuscript fully available?

Reviewer #1: Yes

Reviewer #2: Yes

4. Is the manuscript presented in an intelligible fashion and written in standard English?

Reviewer #1: Yes

Reviewer #2: Yes

5. Review Comments to the Author

Reviewer #1: Very interesting paper on an important topic espeicially at this global context. The paper is well-written with solid qualitative analysis. However I would like to address some problems with it.

1. I see most of the conversation are from the teacher, and only one of them is from a student. I believe it would be beneficial to include more input from the students.

2. I believe the supplementary figures s1, s2 are not addressed. A more detailed discussion of Figures S1 and S2 would enhance the paper's comprehensiveness.

3. I believe including a comparison with schools not implementing SEAs would improve the robustness the argument.

Other than the previous comments, there is one that concerns me the most.. The scope of this research is limited. It's confined within a single secondary school. This could be limiting the generalizability of the findings.

Overall I'm positive about the paper, but I believe it requires some minor revision.

Reviewer #2: As in my view the paper can be accepted , the ethical approval and consent form participants are taken. How ever there needs to be some justification for the sample size and how were the participants selected .some latest software can be used for analysis of the Data.

6. PLOS authors have the option to publish the peer review history of their article (what does this mean?). If published, this will include your full peer review and any attached files.

Reviewer #1: No

Reviewer #2: No

---

## [Author Response · Author response to Decision Letter 0]

26 Feb 2024

Dear Reviewers

a letter has been added to the attach files that breaks down the response to each of your comments. Thank you very much for your input

---

## [Decision Letter · Decision Letter 1]

9 Apr 2024

PONE-D-23-38228R1The effect of Successful Educational Actions in educational transition from primary to secondary school

Dear Dr. Roca,

Thank you for submitting your manuscript to PLOS ONE. After careful consideration, we feel that it has merit but does not fully meet PLOS ONE’s publication criteria as it currently stands. Therefore, we invite you to submit a revised version of the manuscript that addresses the points raised during the review process.Indicate which changes you require for acceptance versus which changes you recommendAddress any conflicts between the reviews so that it's clear which advice the authors should followProvide specific feedback from your evaluation of the manuscriptPlease submit your revised manuscript by May 24 2024 11:59PM .. If you will need more time than this to complete your revisions, please reply to this message or contact the journal office at plosone@plos.org. Please include the following items when submitting your revised manuscript:A rebuttal letter that responds to each point raised by the academic editor and reviewer(s). You should upload this letter as a separate file labeled 'Response to Reviewers'.A marked-up copy of your manuscript that highlights changes made to the original version. You should upload this as a separate file labeled 'Revised Manuscript with Track Changes'.An unmarked version of your revised paper without tracked changes. You should upload this as a separate file labeled 'Manuscript'.We look forward to receiving your revised manuscript.

Kind regards,

Chinaza Uleanya

Academic Editor

PLOS ONE

Journal Requirements:

Reviewers' comments:

Reviewer's Responses to Questions

**Comments to the Author**

1. If the authors have adequately addressed your comments raised in a previous round of review and you feel that this manuscript is now acceptable for publication, you may indicate that here to bypass the “Comments to the Author” section, enter your conflict of interest statement in the “Confidential to Editor” section, and submit your "Accept" recommendation.

Reviewer #2: All comments have been addressed

Reviewer #3: (No Response)

2. Is the manuscript technically sound, and do the data support the conclusions?

Reviewer #2: Yes

Reviewer #3: Partly

3. Has the statistical analysis been performed appropriately and rigorously? 

Reviewer #2: Yes

Reviewer #3: No

4. Have the authors made all data underlying the findings in their manuscript fully available?

Reviewer #2: Yes

Reviewer #3: No

5. Is the manuscript presented in an intelligible fashion and written in standard English?

Reviewer #2: Yes

Reviewer #3: No

6. Review Comments to the Author

Reviewer #2: The paper can be accepted without any further revision. manuscript describes a technically sound piece of scientific research with data that supports the conclusions. . The conclusions are be drawn appropriately based on the data presented. it covers a wide range of information; the results have been presented well. The conclusion is quite comprehensive

Reviewer #3: The topic is interesting and important. The scope of immigration in Europe and the West in general, require the receiving countries to create mechanisms that will benefit the immigrants and at the same time preserve the existing social structure in those countries. Inequality, possible gaps in education undoubtedly threatens the stability of those countries and therefore any program that aims to the reduction of those gaps contribute to both the immigrants and the receiving countries. The program presented in the article is an important program and the attempt to test its effectiveness is noteworthy. However, there are several methodological problems:

1- Literature review – has to be upgraded and refined in reference of the innovation stated in the article. This can be achieved with additional literature review dated from the last two years.

2- Methods - It was specifically stated that the data collection was done through interviews, never-the-less, the article also includes quantitative data. In this case, the research is mix-method research, meaning it combines quantitative and qualitative methods. Therefore, proof and data have to be included in the methodology (method) chapter.

3- Methods - It is advisable to explain in details the combination of the methods and what each method contributes to the understanding and the effectiveness of the program, such as promoting the well-being, sense of belonging and involvement in the school life of immigrant and refugee children.

4- Analysis - In case of analysis, we look to test a change before and after an intervention. Therefore it is not enough to examine the change only according to frequencies, but it is necessary to use tests that include the value of significance. Meaning, without significance tests it is not possible to state any or a "significant" difference.

5- Method - it is stated that the school has about a thousand students, however for this research, as stated in the article, only five students and ten teachers were selected for interviews. It is necessary to increase the number of students interviewed significantly, as well as the number of teachers.

6- It is advisable to add background variables such as ages. It is necessary to add the ages of the students who participated in the study, As the gaps might defer accordingly. Also, table 1 states the number of years of seniority of the teachers, however, no figure represents the data analysis. It is worth adding a reference to the contribution of this variable.

7- How were the respondents sampled in the study? It is necessary to add the information to the method chapter.

8- The article should be proof read prior to publishing. There are several grammatical and language errors.

9- Bibliography – Lack of uniformity in the bibliography. For example, at the beginning of the article the last names of the authors and the year of publication are written, and towards the end of the article the numbers of the articles as they appear in the final list appeared in parentheses. Also, there are unnecessary punctuation marks in some of the sources and in general there is a need for precision in the way of writing.

7. PLOS authors have the option to publish the peer review history of their article (what does this mean?). If published, this will include your full peer review and any attached files.

Reviewer #2: No

Reviewer #3: No

---

## [Author Response · Author response to Decision Letter 1]

29 Apr 2024

We have included a document with the response to each of the reviewers' observations. Thank you very much for improving our work

---

## [Decision Letter · Decision Letter 2]

16 May 2024

The effect of Successful Educational Actions in educational transition from primary to secondary school

PONE-D-23-38228R2

Dear Dr Roca

We’re pleased to inform you that your manuscript has been judged scientifically suitable for publication and will be formally accepted for publication once it meets all outstanding technical requirements.

Kind regards,

Chinaza Uleanya

Academic Editor

PLOS ONE

Additional Editor Comments (optional):

Based on the reports of the two reviewers, the manuscript is accepted for publication.

Congratulations.

Reviewers' comments:

Reviewer's Responses to Questions

**Comments to the Author**

1. If the authors have adequately addressed your comments raised in a previous round of review and you feel that this manuscript is now acceptable for publication, you may indicate that here to bypass the “Comments to the Author” section, enter your conflict of interest statement in the “Confidential to Editor” section, and submit your "Accept" recommendation.

Reviewer #1: All comments have been addressed

Reviewer #2: All comments have been addressed

2. Is the manuscript technically sound, and do the data support the conclusions?

Reviewer #1: Yes

Reviewer #2: Yes

3. Has the statistical analysis been performed appropriately and rigorously? 

Reviewer #1: Yes

Reviewer #2: Yes

4. Have the authors made all data underlying the findings in their manuscript fully available?

Reviewer #1: Yes

Reviewer #2: Yes

5. Is the manuscript presented in an intelligible fashion and written in standard English?

Reviewer #1: (No Response)

Reviewer #2: Yes

6. Review Comments to the Author

Reviewer #1: Many of the concerns are addressed, and it's great that we see some comments from the students. Overall pretty good.

Reviewer #2: the authors have made changes as per the reviewers comments The paper can be accepted without any further revision. manuscript describes a technically sound piece of scientific research with data that supports the conclusions. . The conclusions are be drawn appropriately based on the data presented. it covers a wide range of information; the results have been presented well. The conclusion is quite comprehensive

7. PLOS authors have the option to publish the peer review history of their article (what does this mean?). If published, this will include your full peer review and any attached files.

Reviewer #1: No

Reviewer #2: No

---

## [Editor Report · Acceptance letter]

31 May 2024

PONE-D-23-38228R2 

PLOS ONE

Dear Dr. Roca, 

I'm pleased to inform you that your manuscript has been deemed suitable for publication in PLOS ONE. Congratulations! Your manuscript is now being handed over to our production team.

Kind regards, 

on behalf of

Dr. Chinaza Uleanya 

Academic Editor

PLOS ONE